Recent mobility of plastid encoded group II introns and twintrons in five strains of the unicellular red alga Porphyridium

Perrineau Marie-Mathilde 1
Price Dana C. 1
Mohr Georg 2
Bhattacharya Debashish 1 3 debash.bhattacharya@gmail.com
1 Department of Ecology, Evolution and Natural Resources, Rutgers University , New Brunswick, NJ , USA
2 Institute for Cellular and Molecular Biology, University of Texas at Austin , Austin, TX , USA
3 Department of Marine and Coastal Sciences, Rutgers University , New Brunswick, NJ , USA
Brogna Saverio
Electronic publication date: 2015 Jun 18
Publication date: 2015
Volume: 3
Electronic Location ID: e1017
Received 2014 Dec 19; Accepted 2015 May 22
Copyright: © 2015 Perrineau et al.
Copyright year: 2015
Copyright holder: Perrineau et al.
License: This is an open access article distributed under the terms of the Creative Commons Attribution License, which permits unrestricted use, distribution, reproduction and adaptation in any medium and for any purpose provided that it is properly attributed. For attribution, the original author(s), title, publication source (PeerJ) and either DOI or URL of the article must be cited.
License URL: https://creativecommons.org/licenses/by/4.0/

Keywords: Group II introns, Twintrons, Red algae, Porphyridium, Mobile genetic elements, Plastids

Funding: National Science Foundation 1004213 United States Department of Energy DE-EE0003373/001 NIH GM37949 Welch Foundation F-1607 The work was funded by a grant from the National Science Foundation (1004213) and from the United States Department of Energy (DE-EE0003373/001) awarded to Debashish Bhattacharya. Research by Georg Mohr is supported by NIH grant GM37949 and Welch Foundation grant F-1607 to Alan M. Lambowitz. The funders had no role in study design, data collection and analysis, decision to publish, or preparation of the manuscript.

==============================
Group II introns are closely linked to eukaryote evolution because nuclear spliceosomal introns and the small RNAs associated with the spliceosome are thought to trace their ancient origins to these mobile elements. Therefore, elucidating how group II introns move, and how they lose mobility can potentially shed light on fundamental aspects of eukaryote biology. To this end, we studied five strains of the unicellular red alga Porphyridium purpureum that surprisingly contain 42 group II introns in their plastid genomes. We focused on a subset of these introns that encode mobility-conferring intron-encoded proteins (IEPs) and found them to be distributed among the strains in a lineage-specific manner. The reverse transcriptase and maturase domains were present in all lineages but the DNA endonuclease domain was deleted in vertically inherited introns, demonstrating a key step in the loss of mobility. P. purpureum plastid intron RNAs had a classic group IIB secondary structure despite variability in the DIII and DVI domains. We report for the first time the presence of twintrons (introns-within-introns, derived from the same mobile element) in Rhodophyta. The P. purpureum IEPs and their mobile introns provide a valuable model for the study of mobile retroelements in eukaryotes and offer promise for biotechnological applications.

Introduction

Nuclear genome evolution and eukaryotic cell biology in general are closely tied to the origin and spread of autocatalytic group II (GII) introns. These parasitic genetic elements are thought to have initially entered the eukaryotic domain through primary mitochondrial endosymbiosis (e.g., Rogozin et al., 2012; Doolittle, 2014), and are implicated as a selective force behind formation of the nuclear compartment (Aravind, Iyer & Koonin, 2006; Martin & Koonin, 2006). Ultimately, GII introns were transferred to the nucleus and gave birth to the forerunners of nuclear spliceosomal introns and the small RNAs associated with the spliceosome (Cech, 1986; Sharp, 1991; Qu et al., 2014). This explanation of intron origin, although widely held to be true (e.g., Rogozin et al., 2012) is nonetheless shrouded in the mists of evolutionary time. Understanding more recent cases of GII intron gain and loss are vital to testing ideas about the biology of autocatalytic introns. Here we studied GII intron evolution in five closely related strains of the unicellular red alga Porphyridium purpureum (Rhodophyta) that surprisingly contain over 40 intervening sequences in their plastid genomes (Tajima et al., 2014). Red algae are not only interesting in their own account as a taxonomically rich group of primary producers (Ragan et al., 1994; Bhattacharya et al., 2013) but they also contributed their plastid to a myriad of chlorophyll c-containing algae such as diatoms, haptophytes, and cryptophytes through secondary endosymbiosis (Bhattacharya, Yoon & Hackett, 2004; Archibald, 2009). Therefore, GII introns resident in red algal plastid genomes could also have entered other algal lineages through endosymbiosis.

With these ideas in mind, we explored the genetic diversity, secondary structure, and evolution of GII introns and their mobility-conferring intron-encoded proteins (IEPs; Lambowitz & Zimmerly, 2011) in the plastid genome of five strains of P. purpureum, four of which were determined for this study. Phylogenetic analyses show that the P. purpureum IEPs and their introns are monophyletic, suggesting a shared evolutionary history (Toro & Martínez-Abarca, 2013). Analysis of IEPs reveals key traits associated with GII intron mobility and loss, and analysis of secondary structures uncover unique features of red algal group II introns. We also report for the first time the presence of twintrons (introns-within-introns) in Rhodophyta plastid genomes and deduce their recent origins from existing IEPs that targeted heterologous DNA sites. In summary, our study identifies a promising red algal model for the study of GII intron biology and evolution and suggests these mobile elements could potentially be harnessed for biotechnological applications (Enyeart et al., 2014).

Materials and Methods

Porphyridium purpureum strains and plastid genomes

Four Porphyridium purpureum strains, SAG 1380-1a, SAG 1380-1b, SAG 1380-1d (obtained from the Culture Collection of Algae, Göttingen University) and CCMP 1328 (obtained from the National Center for Marine Algae and Microbiota, East Boothbay, ME) were grown under sterile conditions on Artificial Sea Water (Jones, Speer & Kuyr, 1963) at 25 °C, under continuous light (100 µmol photons m−2 s−1) on a rotary shaker at 100 rpm (Innova 43; New Brunswick Eppendorf, Enfield, Connecticut, USA). Cells were pelleted via centrifugation and DNA was extracted from ca. 100 mg of material with the DNeasy Plant Mini Kit (Qiagen) following the manufacturer’s protocol. Sequencing libraries were prepared for each strain using the Nextera DNA Sample Preparation Kit (Illumina Inc., San Diego, California, USA) and sequenced on an Illumina MiSeq sequencer using a 300-cycle (150 × 150 paired-end) MiSeq Reagent Kit v2 (Illumina, Inc.). Sequencing reads were quality and adapter trimmed (Q limit cutoff = 0.05) and overlapping pairs were merged at the 3′ end using the CLC Genomics Workbench 6.5.1 (CLC Bio, Aarhus, Denmark).

Mapping, polymorphism detection and analysis

The reads from each of the four newly-sequenced strains above were mapped to the existing P. purpureum plastid reference genome (strain NIES 2140; Tajima et al., 2014) with a stringency of 90% sequence identity over a 90% read length fraction using the CLC Genomics Workbench (CLC Bio, Aarhus, DK). SNPs were called using the Genomics Workbench 6.5.1 quality-based variant detection (≥10× base coverage, quality score >30 and ≥50% frequency required to be called). An uncorrected distance phylogeny was constructed using a matrix of DNA polymorphisms detected between the five plastid genomes with the program MEGA6.06 (Tamura et al., 2013; 100 bootstrap replicates).

Group II intron and IEP identification

Novel GII introns in the plastid genomes of the four P. purpureum strains were identified by aligning de novo assembled (using the CLC Genomics Workbench v.6.5.1 de novo assembler) plastid contigs from each strain to the NIES 2140 reference. Multiple large (>50bp) insertions were identified in our de novo contigs with respect to the reference, and were annotated as putative introns. We then mapped the corresponding raw short read data to these contigs and manually inspected the mapping for assembly artifacts. Intron encoded proteins (IEPs) were identified within the putative introns by ORF detection using the bacterial/plastidic genetic code. The four domains that constitute an IEP (i.e., reverse transcriptase [RT], maturase [X], DNA-binding [D], and endonuclease [En] Mohr, Perlman & Lambowitz, 1993; San Filippo & Lambowitz, 2000) were identified by sequence alignment using ClustalX (Larkin et al., 2007) to known IEPs of the prokaryote CL1/CL2 group and to those from the Rhodophyta, Viridiplantae, Cryptophyta, Euglenozoa, and stramenopiles (listed in Table S1) obtained from NCBI and the Group II intron database (Dai et al., 2003; http://webapps2.ucalgary.ca/~groupii/, accessed Sept. 2014). To examine the phylogeny of these mobile elements, the IEP peptide sequences were aligned with the RT-domain alignment of Toro & Martínez-Abarca (2013) and maximum likelihood phylogenies were inferred under the WAG amino acid substitution model with 100 bootstrap replicates using MEGA6.06. The GII intron/IEP sequences described here are accessible using NCBI accession numbers KKJ826367 to KKJ826395 and the P. purpureum plastid genome under NC_023133 (Tajima et al., 2014).

Intron structure and evolution

Intron secondary structures were predicted using sequence alignment, manual domain identification, and automatic structure conformation in comparison with previously predicted structures of group IIB introns using the Mfold web server (Zuker, 2003; Table S1). A detailed secondary structure model was generated based on the rpoC1 intron and mat1d IEP (Fig. 1). This was then used as a guide to predict draft structures using PseudoViewer3 (Byun & Han, 2009) for all other GII introns. A domain alignment was then performed against the GII intron structures derived from the cryptophyte Rhodomonas salina (Maier et al., 1995; Khan et al., 2007) using ClustalX2.1, and a maximum-likelihood phylogeny was generated using intronic nucleotide sequence data under the GTR +I + Γ model with 100 bootstrap replicates using MEGA6.06 (Tajima et al., 2014). Prior to this, the IEPs or IEP remnants were removed to avoid potential long-branch attraction artifacts. Additionally, conserved motifs within the basal DI, DIV, DV and DVI domains (Table S2) were used as a BLASTN (Altschul et al., 1990) query to the five aligned plastid genomes to identify additional group II intron structures present in all strains (and thus not identified via length heterogeneity upon initial assessment).

Figure 1 P. purpureum group IIB intron structure.

Predicted structure of the rpoC1 intron containing the mat1d IEP. The structure is composed of six conserved domains (DI–DVI). Exon and intron binding site (EBS and IBS) and Greek letters indicate nucleotide sequences involved in long-range tertiary interactions. The IEP is located in the DIV domain

The twintrons present in the P. purpureum plastid genome were aligned and compared to the other introns to allow identification of the outer and inner introns, exon binding sites, to describe their secondary structures, and potentially to understand their mode of origin.

Results and Discussion

Paired-end short read sequencing of P. purpureum strains SAG 1380-1a, SAG1380-1b, SAG 1380-1d and CCMP1328 generated 5.5M, 3.4M, 2.7M and 4.3M reads, respectively, after trimming and quality control. These data covered between 98 and 100% of the NIES2140 plastid reference genome (information regarding read mapping and coverage of the reference can be found in Table 1). A phylogenetic tree of the five studied P. purpureum strains inferred on the basis of 332 single nucleotide polymorphisms (SNPs) present in their plastid genomes demonstrates the close evolutionary relationship between the four strains reported here (SAG 1380-1a/b/d, CCMP-1328) with respect to strain NIES 2140 (Fig. 2A; Tajima et al., 2014). By examining length heterogeneity within these plastid genome sequence alignments, we identified four novel GII intron/IEP combinations (mat1f, 1g, 1h, 1i; Table 2 and Fig. 2B) in addition to the five previously reported by Tajima et al. (2014; mat1a, 1b, 1c, 1d, 1e). These novel elements exhibited lineage-specific distributions on the phylogeny, whereas those encoding mat1a, b, c and e were recovered from all strains (Fig. 2B). Using conserved structural motifs (see Fig. S1 and ‘Materials & Methods’) as the basis for a homology search within remaining intronic and intergenic P. purpureum plastid sequence, we defined two additional GII introns (within int mntA, int.a rpoB), and an intergenic element with GII intron structure located between the psbN and psbT genes. Each of the three structures is present in all four strains, and contain remnant (or ‘ghost’) ORFs that have lost their IEPs via sequence degeneration or excision. These structures were subsequently included in our analyses.

Figure 2 Evolution of group II introns and IEPs in Porphyridium strains.

(A) Neighbor-joining phylogenetic tree (uncorrected p-distance, 100 bootstrap replicates, branch supports >70% shown) built using 332 SNPs identified in these plastid genomes. Blue arrows illustrate the distribution of group II introns containing IEPs or IEP remnants described by Tajima et al. (2014); green arrows denote group II introns containing IEPs newly described in this study, and orange arrows illustrate twintrons defined here. (B) Location of group II introns/IEPs (from Fig. 2A) in the plastid genomes (not to scale). Blue arrows illustrate the distribution of group II introns containing IEPs or IEP remnants described by Tajima et al. (2014); green arrows denote group II introns containing IEPs newly described in this study, and orange arrows illustrate twintrons defined here.

Table 1 Porphyridium purpureum plastid sequencing data generated.

Illumina sequencing data generated for each P. purpureum strain referenced in this study.

Strain	Total reads	Trimmed reads	Reads mapped	Ref. length	% Ref covered	Avg. cov.	% Ref ≥10× cov	
SAG 1380-1a	5,665,926	5,539,699	74,904	212,133	97.4	52.87	84	
SAG 1380-1b	3,639,740	3,639,740	72,186	215,863	99.2	40.76	95	
SAG 1380-1d	2,827,948	2,696,004	54,422	215,440	90	35.14	88	
CCMP 1328	4,524,336	4,350,554	83,716	216,010	99.2	77.41	96	

Table 2 Group II introns and associated features.

IEP-containing group II introns from Tajima et al. (2014) (TEA) and IEP or IEP remnant containing group II introns described in this study are listed. Presence of reverse transcriptase (RT), maturase (MAT), endonuclease (En) and YADD motif are noted.

IEP	Reference	Location	IEP present?	RT	MAT	DNA	En	YADD	
mat1a	TEA	intergenic atpB-atpE	YES	YES	YES	TRUNCATED	NO	ISDQ	
mat1b	TEA	int.b dnaK	YES	YES	YES	TRUNCATED	NO	FGNK	
mat1c	TEA	int.c infC	YES	YES	YES	TRUNCATED	NO	YVDD	
mat1d	TEA	int gltB	YES	YES	YES	YES	YES	YADD	
mat1e	TEA	int.a rpoC2	YES	YES	YES	TRUNCATED	NO	YADD	
mat1fa	This study	int.b rpoC2	YES	YES	YES	YES	YES	YADD	
mat1fb	This study	atpI int.b	YES	YES	YES	YES	YES	YADD	
mat1fc	This study	int atpB	YES	YES	YES	YES	YES	YADD	
mat1g	This study	int rpoC1	YES	YES	YES	YES	YES	YADD	
mat1h	This study	int ycf46	YES	YES	YES	YES	YES	YADD	
mat1i	This study	int tsf	YES	YES	YES	YES	YES	YADD	
no IEP	This study	intergenic psbB-psbT	NO (‘GHOST’)	N/A	N/A	N/A	N/A	N/A	
no IEP	This study	int mntA	NO (‘GHOST’)	N/A	N/A	N/A	N/A	N/A	
no IEP	This study	int.a rpoB	NO (‘GHOST’)	N/A	N/A	N/A	N/A	N/A	

We identified six new GII intron insertion sites in our P. purpureum strains encoding the mat1fa, 1fb, 1fc, 1g, 1h, 1i IEPs (see Table 2) in addition to the five sites previously described in the NIES 2140 strain (encoding mat1a, 1b, 1c, 1d, 1e; Tajima et al., 2014; see Fig. 2). Among the nine GII intron/IEP combinations present, only four occur at the same insertion site in all strains (mat1a, 1b, 1c, and 1e), whereas four are unique to individual strains (mat1d, 1g, 1h, and 1i). The mat1fa and mat1fb IEPs are identical at the nucleotide level and form twintrons (see below), whereas mat1fc contains a single SNP.

A maximum-likelihood phylogeny was constructed using an alignment of the novel GII introns described in this study, along with the 42 introns present in NIES 2140 (with IEP sequences removed from the alignment; Fig. 3). This analysis demonstrates that twelve IEP/IEP-remnant containing GII introns in P. purpureum form an exclusive monophyletic group (88% bootstrap support), whereas the mat1a- and mat1b-encoding elements are sister taxa in a distantly related and evolutionarily diverged clade. Despite partial nucleotide sequence conservation (Fig. S1), the intergenic structure encoding mat1a could not be folded into a functional group II intron structure (only domains DIV-DVI could be identified Fig. S2) , and we were unable to identify any group II intron secondary structural homology within the mat1b-encoding intron (see Fig. S1 and the section below entitled, ‘Group IIB intron secondary structure’). These structures may then represent “group II-like introns” as defined by Toro & Nisa-Martínez (2014) in that they lack canonical secondary structures and yet maintain a maturase domain. In addition, the GII intron structures with remnant or ghost ORFs recovered in our analysis formed a monophyletic group with those that maintained functional IEPs. These results are consistent with the evolutionary model widely accepted for group II introns (Toor, Hausner & Zimmerly, 2001; Simon, Kelchner & Zimmerly, 2009) that predicts co-evolution of IEPs and self-splicing RNAs, and suggests that IEP-lacking (remnant) introns derive from introns that once contained a functional mobility-conferring enzyme.

Figure 3 Phylogeny of P. purpureum group II introns.

Maximum likelihood tree; only bootstrap values >70% are shown. To avoid long-branch attraction, the IEP or IEP remnant sequences (indicated in bold) were removed from the alignment. Colored circles indicate presence (blue) or absence (red) of DNA-binding domain, Endonuclease domain and intact YADD motif, respectively.

Intron-encoded proteins

Intron-encoded proteins present at the same insertion site are nearly identical among the strains (98.9–100% amino acid identity), except for the mat1b IEP in strain NIES 2140 which has an apparent truncation of 27 amino acids due to a premature stop-codon. All nine IEPs contain two fully conserved reverse transcriptase (RT) and maturase (X) domains (Fig. S3), whereas four of the five elements present in all five P. purpureum strains (mat1a, 1b, 1c, 1e) are either truncated or have completely lost the DNA-binding (D) and endonuclease (EN) domains responsible for conferring mobility (Simon, Kelchner & Zimmerly, 2009). These latter GII introns thus appear to have lost mobility, and exhibit vertical inheritance. Additionally, mat1a and mat1b lack the YADD motif crucial for reverse transcriptase activity at the active site (Fig. S3; Moran et al., 1995). The remaining five GII introns encoding mat1d, mat1f[a,b,c], mat1g, mat1h, mat1i are distributed in lineage-specific patterns on the P. purpureum phylogeny (Fig. 2A) and likely remain mobile because they retain all functional domains (Fig. S1). Thereforre, we show here for the first time examples of recent intron mobility and putative stability; the latter being represented by plastid-encoded IEPs that lack a functional endonuclease domain due to mutation and/or sequence degeneration.

Phylogenetic analysis using the IEP peptide alignment shows that seven of the nine P. purpureum IEPs form a monophyletic clade that is sister to cryptophyte plastid IEPs, the cyanobacterial CL2B clade, and Euglenozoa plastids (Fig. 4 and Fig. S4). The mat1a and mat1b IEPs, derived from group II introns found to lack typical secondary structure, create a paraphyletic assemblage within the cryptophytes (mat1a) or group outside of the CL2B clade (mat1b). This tree, in association with Fig. 3, illustrates the shared ancestry and subsequent co-evolution of seven IEPs as well as their associated GII intron structures.

Figure 4 Phylogeny of CL2B group II IEPs.

The nine plastid-encoded IEP sequences from P. purpureum were added to selected sequences from the bacterial group II intron database, together with Cryptophyta and Euglenozoa IEPs (ML, bootstrap >70%). The tree is rooted with proteins from the CL2A, CL1A, and CL1B groups (including the mat1b IEP). Note: the mat1f IEP represents the three nearly identical IEP sequences (mat1fa, mat1fb, mat1fc) described in the text. Colored circles indicate presence (blue) or absence (red) of DNA-binding domain, Endonuclease domain and intact YADD motif, respectively.

Group IIB intron secondary structure

Self-splicing group II introns are dependent on a conserved secondary and tertiary RNA structure. These autocatalytic genetic elements are composed of six distinct double-helical domains (DI to DVI) that radiate from a central wheel with each domain having a specific activity (Lambowitz & Zimmerly, 2011). As illustrated by the rpoC1 GII intron that contains mat1d (Fig. 1), the introns studied here have group IIB intron secondary structures following this model. Annotated sequence alignments and draft secondary structures for the remaining introns are presented in the supplementary information (Figs. S5–S16 (note that no intronic sequence data were removed to simplify folding)). As expected, the P. purpureum IEPs are located in the domain IV (DIV) loop, which is integral for ribozyme activity. DIVa (the maturase binding site exclusive of the IEP (see Fig. S23)) and DV contain conserved regions (96 ± 4% identity), whereas DVI is highly variable (37 ± 17% identity; length range 44–162bp; see Fig. S17).

The bulged AC nucleotide pair illustrated within DV of Fig. 1 is in agreement with the model of Toor et al. (2008) and Keating et al. (2010), however the possibility exists that (as in the remaining introns (Figs. S5–S16)) the unpaired nucleotides can be shifted downstream to create a CG bulge. The DVI domain contains a conserved, bulged adenosine that serves as a nucleophile during lariat generation upon splicing (Peebles et al., 1987; Robart et al., 2014), however most P. purpreum group II intron models described here maintain an additional unpaired guanine in an AG bulge. The effect this has on the splicing reaction remains unknown. Structural analysis reveals a novel and unusual bipartite DIII domain configuration, because it can be represented by either a canonical stem/loop structure, or as two individual stems (Figs. S5–S11 (see inset DIII)), or as two individual stems only (Figs. S12–S16). The DIII domain contributes an adenosine pair to a base stack that serves to reinforce DV opposite the catalytic site, and stabilizes the entire structure (Robart et al., 2014). Modification of this domain in the P. purpureum group II intron structures that have lost mobility may reflect the lack of an IEP and thus the need for reinforcement.

Group II intron RNAs self-splice via base-pairing interactions between exon-binding sites (EBS1 & EBS2) on the ribozyme and intron-binding sites (IBS1 & IBS2) at the 5′ exon region (Lambowitz & Zimmerly, 2011). Despite a common origin, the P. purpureum introns that encode an IEP appear to have a highly variable EBS (Fig. S18) perhaps explaining their ability to spread to novel sites in these plastid genomes. Each EBS/IBS pairing is uniquely associated with an intron/IEP combination, and complementarity between both is present. EBS1 and/or EBS2 were not identified for the mat1a, mat1b, and mat1c introns. Interestingly, EBS1 is located at the same site in the nucleotide alignment, whereas the EBS2 position is variable due to length heterogeneity between introns. Understanding how variation in these binding sites affects the ability of group II introns to self-splice and bind target DNA is paramount for ‘targetron’ development (Enyeart et al., 2014) and application of these mobile elements to biotechnology.

Finally, sequence alignment of the P. purpureum introns described here with the five Rhodomonas salina introns presented in Khan & Archibald (2008) (Fig. S17) demonstrates that the domain organization and secondary structure of these elements in both species are similar. We were thus able to derive amended secondary structures for the cryptophyte models proposed by Maier et al. (1995) and Khan & Archibald (2008) using P. purpureum as a guide. In doing so, we identified a cryptophyte domain IVa similar to that in P. purpureum that contains the IEP and has modified domains DII and DIII (e.g., Fig. S19). We propose that the non-canonical features described by Khan & Archibald (2008) in R. salina and H. andersenii (i.e., domain insertions, ORF relocation, absence of internal splicing) can be explained by degeneration of the endonuclease domain between the protein C-terminus and domain IVa. Amended structures for the remaining cryptophyte introns are presented herein (Figs. S19–S23).

Red algal twintrons

Introns nested within other introns (or twintrons) were first reported in the Euglena gracilis plastid (Copertino & Hallick, 1991). Since then, group II/III twintrons have been reported at multiple sites in complete Euglenozoa plastid genomes (E. gracilis and Monomorphina aenigmatica; Pombert et al., 2012) and from the plastid genomes of the cryptophytes Rhodomonas salina and Hemiselmis andersenii (Maier et al., 1995; Khan et al., 2007 (however see discussion, above)). Twintrons have also been described in the prokaryotes Thermosynechococcus elongatus, a thermophilic cyanobacterium (Mohr, Ghanem & Lambowitz, 2010) and in Methanosarcina acetivorans, an archaebacterium (Dai & Zimmerly, 2003). Here we provide the first description of twintrons in rhodophyte plastid genomes, and the first known report of an inner intron (mat1f) found nested within two different outer introns (while also inserted in a third gene). The plastid genomes of three P. purpureum strains each contain two twintrons encoding mat1fa and mat1fb (Figs. 2A and 2B) that are bounded by different outer introns inserted in the rpoC2 and atpI genes, respectively. Two strains contain a copy of the inner intron/IEP inserted singly within the atpB gene as mat1fc. Alignment of the outer and inner twintron regions together with the other introns shows that the two different twintrons have very similar structures (Fig. S1) Despite partial sequence similarity (78.2% sequence identity in pairwise comparisons), the two outer introns have similar IEP remnants. The IEPs are truncated at the same site, likely due to a partial protein deletion. Approximately 130 nt and 555 nt, respectively, remain in the 5′ and 3′ regions of the former IEP in the external introns. Presumably, the later insertion of the inner intron happened at the same binding site (85 nt further downstream from the excision site). Our analyses show that the closely related outer introns int.b (atpI) and int.b (rpoC2; Fig. 3) in P. purpureum retain IEP remnants that have been truncated in the same region due to inner intron insertion at the same DIV target site (Fig. S17). Of future interest is to study the splicing of these red algal twintrons to confirm that excision occurs in consecutive steps as in other plastid twintrons (Copertino, Shigeoka & Hallick, 1992).

Conclusions

In summary, our results support a relatively simple explanation for the origin of a complex family of group II introns in the plastid genome of different P. purpureum strains (see Fig. 2A). We suggest that the common ancestor of these five strains contained several IEP-encoding group II introns that may trace their origin to the cyanobacterial primary plastid endosymbiont. In turn, the Cryptophyta may have acquired these group II introns during the secondary endosymbiosis of a red alga potentially related to a Porphyridium-like donor. These hypotheses require testing with additional plastid genome data from red algae and cryptophytes. Regardless of the time or mode of origin our data suggest that seeds for nuclear spliceosomal introns exist in red algae vis-à-vis organelle encoded group II introns.

It is also clear that during evolution, some mobile group II introns lose their IEP either by complete deletion, partial degeneration (i.e., loss of the YADD motif), or by point mutations that resulted in-frame stop codons (as in the En domain). All of these events create mobility-impaired introns that are stably inherited in descendant lineages. However, some P. purpureum IEPs recovered here have not undergone deleterious change and apparently retain mobility. These mobile introns are inserted in different genes in the plastid genomes, including the intron encoding the mat1f IEP that created two different twintron combinations. We suggest that P. purpureum is a potentially valuable eukaryote model for understanding the evolution of recently mobile group II introns. The presence of active IEPs in the P. purpureum plastid genome also makes this species a good candidate for biotechnological applications, for example via the insertion of IEP encoded foreign genes in plastid genomes (Enyeart et al., 2014). In this regard, P. purpureum synthesizes compounds of interest such as unsaturated fatty acids and photosynthetic pigments (Lang et al., 2011) and plastid transformation is stable, which is rare for red microalgae (Lapidot et al., 2002).

Supplemental Information

Figure S1 Nucleotide alignment of P. purpureum plastid introns

Boundaries used to determine homology are indicated in red (DI stem, DIV stem, DV and DVI stem, respectively). The IEP coding sequences are in yellow. Additional group II introns with degenerate IEPs (i.e., psbN-psbT, int.a rpoB, int mntA, int.b rpoC2) added to analysis are included. The mat1f-encoding group II intron illustrated here represents mat1fc; the nearly identical mat1fa and mat1fb are omitted.

Click here for additional data file.

Figure S2 Draft P. purpureum intron structure (intergenic region between atpB-atpE, mat1a IEP)

Only DIV, DV, and DVI were identified.

Click here for additional data file.

Figure S3 Alignment of P. purpureum intron-encoded protein domains

The four identified domains are separated by an artificial five amino acid gap. The unboxed 5′ sequence comprises the reverse transcriptase (RT) domain. The maturase (X) domain is boxed in black, the DNA-binding (D) domain in red and the endonuclease (En) domain in blue. The D and En domains are partial or absent in four IEPs (mat1a, mat1b, mat1c and mat1e). Asterisks are placed above the YADD domain.

Click here for additional data file.

Figure S4 Phylogeny of CL2B group II intron-encoded proteins

The nine plastidial IEP sequences from P. purpureum were added to selected sequences from the bacterial group II intron database, together with different eukaryote taxa such as Rhodophyta, Cryptophyta, Viridiplantae, Euglenozoa, and stramenopiles from the CL1 and CL2 group. The unrooted tree is annotated with the IEP classes (ML, bootstrap >70%).

Click here for additional data file.

Figure S5 Draft P. purpureum intron structure (int rpoC1, mat1g IEP)

The alternate secondary structure for domain III is depicted in the floating inset.

Click here for additional data file.

Figure S6 Draft P. purpureum intron structure (int tsf, mat1i IEP)

The alternate secondary structure for domain III is depicted in the floating inset.

Click here for additional data file.

Figure S7 Draft P. purpureum intron structure (int ycf46, mat1h IEP)

The alternate secondary structure for domain III is depicted in the floating inset.

Click here for additional data file.

Figure S8 Draft P. purpureum intron structure (int.a rpoC2, mat1e IEP)

The alternate secondary structure for domain III is depicted in the floating inset.

Click here for additional data file.

Figure S9 Draft P. purpureum intron structure (int gltB, mat1d IEP)

The alternate secondary structure for domain III is depicted in the floating inset.

Click here for additional data file.

Figure S10 Draft P. purpureum intron structure (int atpB, mat1f IEP)

The alternate secondary structure for domain III is depicted in the floating inset.

Click here for additional data file.

Figure S11 Draft P. purpureum intron structure (int.b atpI, ORF remnant and outer twintron)

The alternate secondary structure for domain III is depicted in the floating inset.

Click here for additional data file.

Figure S12 Draft P. purpureum intron structure (int.b rpoC2, IEP remnant and outer twintron)

Click here for additional data file.

Figure S13 Draft P. purpureum intron structure (int.c infC, mat1c IEP)

Click here for additional data file.

Figure S14 Draft P. purpureum intron structure (intergene psbN-psbT, IEP remnant)

Click here for additional data file.

Figure S15 Draft P. purpureum intron structure (int.a rpoB, IEP remnant)

Click here for additional data file.

Figure S16 Draft P. purpureum intron structure (int mntA, IEP remnant)

Click here for additional data file.

Figure S17 P. purpureum group II intron/IEP alignment

Alignment of 14 P. purpureum intron/intergenic regions containing an IEP/IEP remnant and four Rhodomonas salina introns. Secondary structures from each domain (DI–DVI) are marked and represented by different colors. The dnaK intron (containing mat1b) does not retain a group IIB intron structure. A partial structure was determined for the atpB-atpEintergenic region (containing mat1a). All the IEPs or IEP remnants are located in domain IV, including the R. salina introns (previously described as the only case of group II intron IEPs located outside of DIV). Twintron insertion sites are indicated with asterisks. The mat1f-encoding structure illustrated here is that encoding mat1fc (int.atpB); the nearly identical mat1fa- and mat1fb-encoding group II introns are omitted.

Click here for additional data file.

Figure S18 Nucleotide alignment of the exon and intron binding sites

The P. purpureum EBS and IBS pairings are unique to each intron/IEP. The complementarity between both is generally preserved; if not, the mutation is located in the 5′ region. EBS1 and/or EBS2 were not identified for the mat1a, mat1b, and mat1c introns. “Ghost” refers to remnant IEPs.

Click here for additional data file.

Figure S19 Modified Rhodomonas salina group II intron secondary structure (groEL gene, strain CCMP 1178)

The domains II, III and IV were modified on the original structure designed by Khan et al. (2007).

Click here for additional data file.

Figure S20 Modified Rhodomonas salina group II intron secondary structure (intron 1, groEL gene, strain CCMP 2045)

The domains II, III and IV were modified on the original structure designed by Khan et al. (2007).

Click here for additional data file.

Figure S21 Modified Rhodomonas salina group II intron secondary structure (intron 2, groEL gene, strain CCMP 2045)

The domains III and IV were modified on the original structure designed by Khan et al. (2007).

Click here for additional data file.

Figure S22 Modified Rhodomonas salina group II intron secondary structure (psbN gene, strain CCMP 1319)

The domains I, II, III, IV and VI were modified on the original structure designed by Maier et al. (1995).

Click here for additional data file.

Figure S23 Modified Rhodomonas salinagroup II intron secondary structure (groEL gene, strain Maier)

The domains I, II, III, IV and VI were modified on the original structure designed by Maier et al. (1995).

Click here for additional data file.

Figure S24 Domain IV primary binding site

The binding sites of the maturases were determined by comparing sequence alignments. The stem-loop structure from a purine-rich internal loop is framed in white, whereas the start- codon is framed in black.

Click here for additional data file.

Table S1 Group II introns used in analysis

Sequences used to guide secondary structure homology search and included in phylogenetic analyses of P. purpureum group II introns.

Click here for additional data file.

Table S2 Query sequences used for structural homology identification

Query sequences used to identify the DI, DIV, DV and DVI domains via BLASTn (Altschul et al., 1990).

Click here for additional data file.

We thank Nicolas Toro for sharing his RT domain-based IEP protein alignment. We are grateful to members of the Genome Cooperative at the Rutgers School of Environmental and Biological Sciences for supporting this research. The authors have no conflict of interest with respect to this work.

Additional Information and Declarations

Competing Interests

Author Contributions

DNA Deposition

The authors declare there are no competing interests.

Marie-Mathilde Perrineau conceived and designed the experiments, performed the experiments, analyzed the data, wrote the paper, prepared figures and/or tables, reviewed drafts of the paper.

Dana C. Price performed the experiments, analyzed the data, wrote the paper, prepared figures and/or tables, reviewed drafts of the paper.

Georg Mohr analyzed the data, contributed reagents/materials/analysis tools, reviewed drafts of the paper.

Debashish Bhattacharya conceived and designed the experiments, performed the experiments, analyzed the data, contributed reagents/materials/analysis tools, wrote the paper, reviewed drafts of the paper.

The following information was supplied regarding the deposition of DNA sequences:

The group II intron/IEP sequences described here are accessible via GenBank accession numbers KJ826367 to KJ826395.

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
