# Peer review of "Recent mobility of plastid encoded group II introns and twintrons in five strains of the unicellular red alga Porphyridium"

_PeerJ, doi:10.7717/peerj.1017_

## Round 0.1 · original submission · Major Revisions

Hi

sorry for the little delay but it was a bit difficult finding suitable reviewers at this time. Both reviewers like the paper but suggest a number of changes which I suggest you address in a revised version of the manuscript.

Reviewer 1 ·

Basic reporting

No commens

Experimental design

No Comments

Validity of the findings

No comments

Additional comments

This manuscript by Perrineau et al describes the identification of novel group II introns in red alga Porphyridium by using highthroughput-sequencing illumina of 4 different strains. They performed phylogenetic analysis of the identified IEPs and intron ribozyme nucleotide sequences. They describe the presence of twinintrons and potential intron RNA structures are also provided. Main issue of this paper is the identification of GII introns that are distributed in a lineage-specific manner in some strains of Porphyridum, and the possibility that some of them could be mobile, which in turn would be of great value for further studies based on these eukaryotic GII and their possible biotechnological applications. This manuscript is an interesting piece of work, but I suggest some revision mainly to improve data presentation.

Comments:
1. ln 34. Authors state that GII introns presumably migrate to the nucleus, but there are also hypothesis that the invasion of the primitive eukaryotic cell triggered eukaryotic compartmentalization separating nucleus from cytoplasm. Perhaps the authors may introduce some reference here.
2. Ln 79. About the illumina sequencing it is not clear from the text if they sequenced again the DNA from the reference strain NIES2140, because it may occur some discrepancies with the published sequence that in turn would be used to establish the cutoff for calling SNPs for the other sequenced strains instead of using 50% frequency as an arbitrary cutoff value. In addition, perhaps they could clarify a little bit more the identification of the insertions as GII intron insertions (Lns 85 to 89).
1. Ln 96. Authors should provide the web page and reference for the group II intron database.
2. Fig 1. In the shown structure at DV they indicated a bulged AC instead of the CG? Please revised according to predicted structures, the lower stem in the figure has 10 pairings and usually there is 9. See also that there is not a legend for the other structures shown in supplementary figures and thus it is no know to what intron they correspond. In any case, I feel that those supplementary figures are not really necessary. The K nucleotides in DI should be indicated.
3. Especially interesting is the bulged GA in DVI (Fig. 1). I have seem similar features in some introns in the database, but the authors could discuss a little bit more this feature and the possibility of some impairment in the splicing reaction if known.
4. Authors state different ways to identify the novel introns/IEP combinations in the P. purpureum plastid genome. By length heterogeneity they identified 4 (mat1f to 1i) and three others using a conserved structural motif (int mntA, int.a rpoB, and an intergenic psbN-psbT). They also seem to identify the encoded ORFs (RT domain?), when present, but the description in the materials and methods section of and in the first part of the results section about the identification procedure is a little bit confusing, perhaps they could rewrite these sections to be more clear on the identification procedure and indicate more clearly (lns 111 and 112) the consensus sequence for the conserved motifs (DI, DIV, DV and DVI) used for the identification, which is not obvious from FigS1.
5. The nomenclature used for the intergenic introns is not obvious, why are they so-called int, int.a and intergenic? Instead of following the mat nomenclature using i.e., mat1j, k and l?. Are they described in table S1?
3. In lns 132 to 138 they describe insertion sites given a nomenclature, but it is not clear if they are actually talking about copy numbers of particular introns i.e the authors would like to say that mat1f is present as two copies (mat1fa and mat1fb)?. Perhaps, this part could be simplified by generating a table showing the main features.
4. Ln 143. What is the meaning of “evolutionary distinct” used for mat1a and mat1b.
5. If mat1a has not a clear GII intron RNA structure (only DIV to VI?) lns 144-145, was it used in the phylogenetic analysis? or not?. The same, if there is not clear secondary structure for a typical GII for mat1b ( lns 145 and 147) was it used in the phylogenetic analysis?.
6. It is also not clear to me the relationships of the 7+5 (12) introns described in the text lns 123 to 131 and the 42 introns that apparently were described in NIES2140. Are the named mat1a to mat1e within these previously 42 introns?. Please clarify; I feel that readers will appreciate to have a clear picture without going to the Tajima paper.
7. The final statement lns 153 to 155 seem to be unrelated with the text above. Please revise.
8. The first part of the intron-encoded proteins, I think it is betters to summarize such information in a table (lns 157 to 166).
9. Identify specifically the names of the five GII that show a lineage specific pattern in the text ln 166.
10. Since the mat1a and mat1b are not associated to a clear GII ribozyme secondary structure (lns 172 and 173), how the authors are sure they are really GII RTs?, there are RTs classified as GII-like (see Simon and Zimmerly 2008 and Toro and Nisa 2014). In addition, if there are not EBS1 and 2 identified how they can be sure of the IBS sequences? See S17 figure.

Reviewer 2 ·

Basic reporting

No Comments

Experimental design

No Comments

Validity of the findings

No Comments

Additional comments

Perrineau et al. sequenced four plastid genomes of different strains of the red algae Porphyridium purpureum. These sequences together with an already available reference genome enabled them to conduct comparative genomics analyses with a special focus on the evolution of plastid intron variability. Interestingly they found that a recently propagated intron is inserted into existing intron sequences and builds two twintron arrangements in the red algae genomes.

The paper is well written and merits publication in PeerJ. However, I have some comments the authors should consider to improve the presentation and clarity of the paper:

Major comments:

The manuscript and all figures follow a complicated intron naming system. For organellar introns as well as their maturases (or IEPs), two nomenclatures are already proposed which I strongly recommend the authors to follow: the suggestions of Dombrovska and Qiu (2004) and Guo and Mower (2013). Adopting both nomenclatures will avoid multiple errors throughout the manuscript where IEP names were used to describe introns (e.g. ‘mat1c is a novel insertion in the atpB gene’, line 137 or ‘where two introns (mat1a and mat1b)’, line 143) and will improve clarity of figures and tables. In addition, following the existing nomenclature would make it unnecessary to include a description of their intron/IEP nomenclature in the methods or results section--which is currently missing.

Minor comments:

Please provide the size of fragments of the sequenced library (lines 68-71) and how many reads were generated per strain.

Can the authors provide more detail on the results of the sequence mapping and de-novo assembly? What was the coverage of the mapped sequences? Was the complete plastid genome covered? Was the final plastid sequence of each strain (with new introns) again verified by read mapping and continuously covered? Depending on the mapping quality, it may be worth mentioning how many positions of the reference genome were excluded from the SNP calling (line 78) since they did not exceed the coverage threshold.

At several parts in the manuscript the authors speak of ‘remnant or ghost ORFs’ (e.g. line 148). Please specify if these ORFs are still intact or represent remnants of former ORFs which are now fragmented and frame-shifted.

Supplementary figure S23 is not mentioned in the manuscript text.

Please specify the difference of domain DIV and DIVa (as mentioned in line 182).

Figures 3 and 4 are only described and discussed separately. A discussion that compares the intron phylogeny in figure 3 with the IEP phylogeny in figure 4 could be added. I would have expected a similar topology of both trees. What is the authors’ opinion on conflicts in the resulting topologies using either intron or IEP sequences (e.g. intron/IEP of mat1c and mat1e)?

Figures:

In the legend of figure 2 it should be stated more explicitly that it only contains a subset of introns which still harbor intact IEPs or in which remnants of IEPs were detected. (If all ‘group II introns described by Tajjima et al. (2014)’ or all ‘IEP-lacking’ introns are included I would expect to count >40 introns as mentioned in line 41.)

It is described that the authors calculated 100 bootstrap replicates in the methods section (line 81) and figure 2A legend but bootstrapping results are neither mentioned in the manuscript nor attached to the tree in figure 2A.

Figure 2B is confusing. Adopting the intron and maturase nomenclature will also help to improve clarity here. In addition, the figure should be drawn to a scale so that arrows are not overlapping. Especially introns ‘mat1fa’ and ‘mat1e’ that are on top of each other could easily be confuse with a twintron arrangement. Moreover, intron ‘mat1fa’ should cause a gap in sequences SAG 1380-1d and NIES 2140.

It would help to add the almost identical sequences of the propagated intron and IEP (forming twintrons and a novel intron in atpB) or to note that ‘int (atpB)’ represents all of these sequences in the phylogeny presented in figure 4, and in the schematics of supplementary figures S1 and S4.

It would be helpful to include the IEP domain structure in figures 4 and 5 to trace the loss of domains during intron or IEP evolution.

The outgroup of the tree in figure 4 should not be collapsed. To see the sequence divergence and because of potential long branch attraction artifacts it would be of interest to see the branch lengths leading to ‘mat1b’ that clusters in the outgroup.

In supplementary table S1 mat1fa, b, and c are named mat1f1, 2, and 3, respectively.

It should be mentioned for supplementary figures S2, S5-S16, S18-S22 if intron regions were left out to simplify the folding. In addition, it would be helpful to indicate at which positions in the intron secondary structures the IEP (or remnants of IEP) are located and where in outer twintrons second introns are inserted.

---

## Round 0.2 · accepted · Accept

I find you have addressed most of the reviewers' criticism. I like the manuscript and I am happy for it to be published.